# Translating Human and Animal Model Studies to Dogs’ and Cats’ Veterinary Care: Beta-Glucans Application for Skin Disease, Osteoarthritis, and Inflammatory Bowel Disease Management

**DOI:** 10.3390/microorganisms12061071

**Published:** 2024-05-25

**Authors:** Andressa Rodrigues Amaral, Larissa Wünsche Risolia, Mariana Fragoso Rentas, Pedro Henrique Marchi, Júlio Cesar de Carvalho Balieiro, Thiago Henrique Annibale Vendramini, Marcio Antonio Brunetto

**Affiliations:** 1Veterinary Nutrology Service, Veterinary Teaching Hospital, School of Veterinary Medicine and Animal Science, University of Sao Paulo, Sao Paulo 05508-270, Brazil; andressa.rodrigues.amaral@usp.br; 2Pet Nutrology Research Center (CEPEN-PET), Department of Animal Nutrition and Production, School of Veterinary Medicine and Animal Science, University of Sao Paulo, Pirassununga 13635-000, Brazil; larissa.risolia@gmail.com (L.W.R.); mvmarirentas@gmail.com (M.F.R.); pedro.henrique.marchi@usp.br (P.H.M.); balieiro@usp.br (J.C.d.C.B.)

**Keywords:** atopic dermatitis, canine, feline, inflammatory bowel disease, osteoarthritis

## Abstract

The inclusion of beta-glucans in dog and cat food is associated with numerous beneficial effects on the health of these animals. In this regard, there is an effort to elucidate the potential of this nutraceutical in chronic patients. Since there is a lack of a review on the topic, this review article aims to compile and discuss the evidence found to date. Atopic dermatitis, inflammatory bowel disease, and osteoarthritis are diseases of significant clinical relevance in dogs and cats. In general, the pathophysiology of these chronic conditions is related to immune-mediated and inflammatory mechanisms. Therefore, the immunomodulation and anti-inflammatory effects of beta-glucans are highlighted throughout this review. The available information seems to indicate that the studies on beta-glucans’ impact on allergic processes in dogs indicate a reduction in clinical signs in atopic dermatitis cases. Additionally, while beta-glucans show promise as a safe supplement, particularly for osteoarthritis, further clinical trials are imperative, especially in uncontrolled environments. Beta-glucans emerge as a potential nutraceutical offering immune benefits for inflammatory bowel disease patients, although extensive research is required to define its optimal origin, molecular weight, dosage, and specific applications across animals suffering from this disease.

## 1. Introduction

Beta-glucans are a type of polysaccharide found within the structure of yeasts, mushrooms, algae, bacteria, barley, and oat [1]. These fibers exhibit a characteristic known as structure–activity relationship, which allows them to perform different biological functions depending on their molecular structure [2]. Generally, all beta-glucans have a central polymeric chain composed of 1,3-beta-glycosidic bonds [3]. However, depending on the source from which they are extracted, they present different patterns and degrees of branching, solubility in water, molecular weight, and structural formation and conformation and effects [4].

So far, it is known that beta-glucans from cereals have linear (1→3) and (1→4) linkages, distinguishing them from non-cereal sources. Those from yeasts are composed of (1→3) and (1→6) linkages with or without branching, while those originating from algae and bacteria are structured in linear (1→3) linkages [5,6]. Among the sources studied, structural differences influence their respective effects on metabolism, which comprise immunomodulatory effects, lipid and glucose metabolism control, antioxidant and antitumor activity, and intestinal microbiome modulation [7,8].

The immunomodulatory effect is better achieved by yeast beta-1,3/1,6-glucans since they are recognized by specific receptors, such as dectin-1, expressed by cells from myeloid origin [8]. Thus, they can stimulate both innate and adaptive immunity by activating neutrophils, B cells, T cells, NK cells, and primarily, macrophages [9]. In general, it is known that beta-1,3/1,6-glucan can stimulate or inhibit the release of cytokines by macrophages and modulate their phagocytic activity [10]. In healthy dogs, studies carried out with the purified beta-1,3/1,6-glucan inclusion in the diet demonstrated favorable effects on glucose regulation [11], immune function [12,13], and in the fecal microbiota [14]. Furthermore, a study conducted by Ferreira et al. (2022) [15] demonstrated that obese dogs supplemented with beta-1,3/1,6-glucans had lower plasma basal glycemia and reduced serum cholesterol, triglyceride, and TNF-alpha levels. Then, its effect on underlying immunocompromised chronic diseases such as atopic dermatitis, osteoarthritis, and inflammatory bowel disease is an interesting object of study.

These chronic diseases are widely represented in the veterinary routine. Atopic dermatitis is the second most prevalent dermatitis in dogs [16]; osteoarthritis is found in 80% of dogs over eight years of age [17]; and inflammatory bowel disease is the most common chronic gastrointestinal disease in dogs and cats [18]. Despite the growing number of research about the positive immunomodulatory effects of beta-glucans, many of those have conflicting results and methodology biases such as the absence of the treatment individualization or homogeneity of experimental groups. Therefore, a review is needed to better understand their applicability and overall quality of evidence.

## 2. Method of Search

The study research on beta-glucans’ implications in skin diseases, osteoarthritis, and inflammatory bowel disease involved extracting from the Embase and PubMed databases. Figure 1 illustrates research available in dogs and cats and their main results. The search covered years from 2020 to 2024, exclusively considering research articles. After review, due to the lack of studies in the interest species (dogs and cats), 3 case reports were added to the skin diseases topic.

The specific terms employed for each topic, the volume of articles retrieved, and the rationale for excluding certain entries, encompassing discrepancies with the subject matter, duplication, and other criteria, are described in Table 1. Articles exclusively utilized for literature review and exploration of mechanisms of action are excluded from the count.

## 3. Beta-Glucans for the Skin

Atopic dermatitis (AD) is one of the most common inflammatory skin diseases and impacts the quality of life of affected individuals. It is the second most diagnosed allergic dermatitis in dogs, after flea bites [16]. The mechanisms by which AD develops have not been completely elucidated; however, from an immunological point of view, this disease is due to the imbalance of Th1/Th2 cells that occurs by inhibiting the proliferation and differentiation of Th1 cells together with the release of cytokines Th2 [19]. Th1 lymphocytes are related to the induction of immune tolerance and cellular immunity reactions. They produce and secrete IL-2, IFN-γ, and TNF-β. Meanwhile, Th2 lymphocytes contribute to the development and persistence of allergic inflammation by producing interleukins [4,5,6,10,13] and activating humoral immunity [20].

Treatment aims to minimize clinical signs and improve quality of life, both in humans and in dogs or cats. Treatment includes blocking or reducing the production of specific interleukins, such as IL-31, IL-4, and IL-5, produced by Th2 lymphocytes through immunosuppressant medications (such as Cyclosporin), monoclonal antibodies (Oclacitinib) or other biologics such as the ones used for other allergies as asthma or rhinosinusitis [21,22], or biological modifiers.

Beta-glucans have important immunomodulatory and anti-inflammatory effects and, therefore, have been tested in an oral and topical way to control this disease. This substance is part a group called biologically active polysaccharides (BAPs) that are modifiers of biological reactions with immunomodulatory activity, both in an immunostimulant and in immunosuppression pattern [23] according to the environment. The main means of action of beta-glucans is mediated by the Dectin-1 receptor, expressed in several immunocompetent cells such as dendritic cells, neutrophils, eosinophils, macrophages, monocytes, and T lymphocytes. Once linked to this receptor, beta-glucans stimulate the production of several cytokines and other mechanisms of immune or non-immune reactions [20].

The effect of BAPs depends on the route of application and other characteristics, such as source, solubility, size of the molecules, and purity of the product [24]. Different types of beta-glucans can also have different mechanisms of action. Another possible mechanism accepted for a type of beta-glucan in the treatment of allergic diseases is the antioxidant capacity through the neutralization of hydroxyl and superoxide radicals. This antioxidant effect occurs through the inhibition of nitric oxide synthase and cyclooxygenase [25]. There are also studies in the literature that relate immune modulation through Th1-type responses [26,27]. Murata et al. (2002) [28] found that the intraperitoneal application of a type of beta-glucan increased the ability of macrophages to produce IL-12, which directed the immune response to the Th1 stimulation.

The routes of application of beta-glucans that have shown effects on allergies can be topical and oral. In a study developed by Jesenak et al. (2016) [29] on 80 humans with AD, beta-glucan was applied topically to the skin. The authors observed a reduction in the duration of exacerbated symptoms of AD associated with a reduction in the severity of injuries in the region of application. In addition, even in these regions there was a reduction in pruritus and in the Eczema Area and Severity Index, indicating the potential use of beta-glucans for this condition.

Kim et al. (2018) [19] used *Aureobasidium pullulans* SM-2001 as a beta-glucan source, which is composed of beta-1,3 and beta-1,6 bonds, the same composition as the yeast wall beta-glucan derivative. In this study, the authors induced AD phenotypes in rats and administered the supplement orally. The signs of itching, histamine secretion, and ear thickness were attenuated in addition to the increase in the production of factors that indicate reactions mediated by Th1 and attenuation of factors that indicate reactions mediated by Th2.

This same research group published another study that evaluated the oral administration of yeast beta-glucans associated or not with the probiotic (*L. plantarum*) in rats with symptoms of AD [30]. The authors observed that the use of beta-glucans associated or not with the probiotic caused a reduction in vasodilation, pruritus, edema, and serum histamine in animals. The authors also found evidence of increased immunomodulation. These effects were associated with the reduction in IgE caused by supplementation of the additives. Other mechanisms have been proposed to elucidate the control of AD symptoms, such as the inhibition of Th2 cellular responses, activation of Treg cells and galectin 9, and the increase in the expression of filaggrin and in the expression of TSLP and bacterial modulation. These effects are based on the fact that Th2 cells secrete cytokines that promote the activation of allergy mediators, and that it is important to inhibit this type of response by stimulating Th1 cells. Galectin-9 is produced and activated by immune cells that increase Th1 responses. Filaggrin is a protein that, when reduced, promotes increased secretion of Th2 cytokines and is associated with increased IgE and disorders of skin barrier functionality. TSLP is a cytokine that expresses keratinocytes from various tissues, in addition to stimulating mast cells and NK-T to cause more Th2 immune responses [31], and the authors noted that treatment was effective in reducing it. The modulation of intestinal bacteria increases the production of galectin 9, which increases Th1 cellular responses in addition to increasing the production of butyrate. This short-chain fatty acid is known to have immunoregulatory effects by stimulating Treg cells (which induce tolerance to harmless antigens) and by increasing the production of IL-10 (cytokine related to immunity inhibition) [32,33]. Finally, the authors highlighted another known mechanism by which beta-glucans can act in immunomodulation, through the mediation of pathogen-associated molecular patterns (PAMPs), such as dectin-1, Toll-like receptors (TLR)-2, TLR-4, and TLR-6, and complement receptor 3, which promote stimuli for cytokine secretion [34,35].

In food allergies, a study has evaluated an inclusion of 0.5 and 1.0% beta-glucans in the diet of rats allergic to ova-albumin (OVA), and a reduction in OVA-specific IgE was observed in comparison to animals who ate the control diet. In addition, supplementation with beta-glucans inhibited the reduction in the production of IL-12 and IFN-gamma in splenocytes [36].

There are not many studies in the literature evaluating the effect of the use of beta-glucans in skin diseases in dogs. Beynen et al. (2011) [18] developed a double-blind, placebo-controlled trial study, with the same type of beta-glucan. For 8 weeks, dogs of several breeds received complete dry food with/without 800 ppm of beta-glucans (16 dogs in control group and 15 dogs in test group) and their clinical signs were evaluated by the owners. The evaluated signs were pruritis, redness, scaling, thickening, and stripping of skin. Coat quality, body condition, and feces consistency were also scored. The signs were scored on day 0 (start) and weekly afterwards. The results were shown by improvement of the sign, calculated by the difference in score between the beginning and the end of the experiment. The test animals had higher group mean scores than control group dogs for the evaluated signs, although those were not significant differences. The authors believe that a placebo effect may have occurred in the control group, which could have underestimated the results for the test group. When compared to baseline values, all signs showed higher scores for both groups. Moreover, as a result, the differences from each sign were summed and the test dogs presented significant improvements compared to the control dogs. The extra improvement caused by the ingestion of beta-1,3/1,6-glucans was 63%. The authors also estimated a dose–response curve, where y values represented the clinical signs scores (ranging from 0–100) and *x* values represented the dose. It was calculated to correct the differences in baseline scores. For a *x* value of 100, a *y* value (response) of 50 was assumed and they then created the following equation: y = 100/(1 + 10^2^ − logx). It was found that the dose equivalents for the combined placebo and treatment effects seen in the test group were much greater than those required for the placebo effect in the control group, which may favor the beta-glucan effect. As a conclusion, the authors stated that the consumption of beta-1,3/1,6-glucans diminishes the clinical signs in dogs with atopic dermatitis. They also observed that beta-glucans are heat stable and can be added to dog food prior to extrusion, as was used in their study.

There was another study that verified the development of atopy with exposure to dust mite allergens, bacterial endotoxins, and fungal glucans. In this study, the relationship between exposure of these possible allergens in the environment and the development of atopy was measured. For this, 28 Labradors with AD and 65 healthy ones were used and samples of dust from the floor of the room, the dogs’ fur, and the dust samples from the place where they slept were collected. From dust samples, concentrations of dust mite allergens, bacterial endotoxins, and beta-1,3-glucan were identified. The authors observed that the concentration of endotoxins in the environment and in animals was inversely proportional to the development of AD, but it did not occur with beta-glucans [37]. This result indicates that the beta-glucan route of administration is important in the regulatory control of AD and that it does not work when it is in the environment.

Other studies in dogs have evaluated the promising effects of beta-glucans for skin diseases, despite not being a single treatment, but part of a major protocol. A case report published by Beyazit et al. (2010) [38] described an eight-year-old German shepherd dog with recurring skin lesions caused mainly by opportunist bacteria and fungi due to demodicosis. This dog was referred to a specialist due to previous refractory treatment (amoxicillin-clavulanic acid at 100 mg/day for four days and ketoconazole at 400 mg/day for two months, then another treatment with amikacin sulfate at 500 mg/day for 10 days, terbinafine hydrochloride at 250 mg/day for two months). All previous treatments were not able to bring skin scraping negative and, after a few months, the infection returned. The dog received ivermectin subcutaneously (0.01 mL/kg/day) for 2 months, enrofloxacin and clavulanic acid-amoxicillin trihydrate were administered intramuscularly for 14 days, and as an immunosupportive measure, 2 capsules (20 mg) of beta-glucan were given once daily. At the end of the first month, a reduction in Demodex spp. count was observed, and clinical and microscopic improvement was noted by the end of the second month. The authors do not mention the beta-glucan’s source and it is notable that the outcome was not necessarily attributed to beta-glucan once the baseline treatment also changed, and the same limitation could be applied for the following studies.

Beta-glucans were also applied in a case of refractory sporotrichosis. A dog with severe destruction of the nasal planum received antibiotic therapy (cephalexin 30 mg/kg) twice a day for 30 days and then was diagnosed with sporotrichosis and treated with itraconazole 10 mg/kg every 12 h. The lesion improved after 3 months; however, the dog still had a positive culture after 7 months. A new approach was taken with beta-1,3-glucan injected subcutaneously every 7 days for 4 weeks at a dose of 0.5 mg/animal and, after that, the culture of the lesion became negative. These authors also did not reveal the beta-glucan’s source [39].

As mentioned before, according to the source and molecule, beta-glucan can be applied topically through the skin when in a soluble form, as a spray. Kim et al. (2023) [40] applied a mixture of beta-glucan, oat lipids, oat peptides, oat flavonoids (phenolic structure), avenanthramides, tocopherol (vitamin E), and sphingomyelinase in the skin of six dogs with atopic dermatitis for 30 days. A pruritus scale and degree of skin lesions was assessed (CADESI-4) and no concurrent treatment was performed with exception of one dog who received a monoclonal antibody due to the severity of the lesions and pruritus. Both scores decreased after treatment and no side effects were observed.

As for cats, beta-glucan was also used topically (gel containing beta-glucan, chlorhexidine digluconate, and pure bee honey) to the wound once daily and orally (a mixture of beta-1,3/1,6-glucan 10 mg/mL, and ascorbic acid and zinc) once a day. The extensive lesion was healed within 6 weeks and topic application was suspended, but oral intake was maintained for 4 more weeks until the beginning of hair growth, and the healing process was shortened by 30% according to the authors [41].

## 4. Beta-Glucans and Osteoarthritis

Orthopedic diseases, especially articular diseases, are very common in the clinical routine of veterinary hospitals and affects animals of any sex or age [17]. Osteoarthritis (OA) is the joint disease most diagnosed in human and veterinary medicine [42]. It is a multifactorial disease and can affect any joint. Its most recent definition is as a joint repair process, which is developed against degeneration and destruction and can cause various biochemical and morphological changes in the joint capsule and synovial membrane, against joint cartilage erosion, osteophyte hypertrophy of bones in joint extremities, and subchondral sclerosis [43]. It can be characterized as primary when it does not have a known and diagnosed cause, or as secondary to abnormalities of the joints in question, which will stimulate the subsequent development of osteoarthritis [44].

The most common predisposing factors in small animals for the development of this condition are humerus-radio-ulnar and hip dysplasia, orthopedic surgery, rupture of the cranial cruciate ligament, joint fractures, and incongruencies resulting from trauma or angular deformities of the limbs [42]. In addition, several risk factors have been reported for osteoarthritis, suggesting that certain systemic and local factors (factors at the animal level, such as race, age, sex, size, and obesity) can considerably affect the development of osteoarthritis [17,42]. The prevalence of osteoarthritis in dogs is reported in the literature with conflicting values. Estimates ranged from 6.6% based on data from O’Neill et al. (2014) [45] to 20% based on data from Pettit et al. (2016) [46] in the United Kingdom canine population. In the United States of America, in the Johnston study, 1997 show age-specific prevalence values ranging from 20% in dogs over one year to 80% in dogs over eight years, based on radiographic and clinical data from Anderson et al. (2018) [17].

The affected joints undergo a degenerative process and are characterized by rupture and loss of articular cartilage, which lead to a decrease in the joint space and friction between the bones, causing swelling, chronic pain, functional impairment, deformity, and consequently, inability to perform functions [47]. An animal with this condition presents signs of pain, lameness, functional disability, and reduced quality of life [48]. Osteoarthritis has no cure; however, surgical treatment can be instituted to correct the primary cause. Therefore, the main recommendation is to control clinical signs [49]. Thus, the prescription of non-steroidal anti-inflammatory drugs (NSAIDs) orally and injection of corticosteroids are commonly used in the clinic in order to reduce pain and inflammation [48]. Unfortunately, the prolonged use of anti-inflammatories is associated with harmful effects [50]. Therefore, there is a demand to find alternative treatments for pain, without causing major damage in the long term. In this context, beta-glucans have been studied [50,51].

The anti-inflammatory effect of beta-glucan occurs through the regulation of inflammatory cytokines, such as nitric oxide (NO), interleukins (ILs), tumor necrosis factor alpha (TNF-α), gamma interferon (INF)-γ, as well as a non-mediator of cytokines, prostaglandin E2 (PGE2). However, few studies have evaluated the anti-inflammatory effects of fungi beta-glucans in animals and humans [51].

Some studies have used beta-glucan as an arthritis inducer in rats, in order to improve the animal model and study possible treatments in humans [52]. That is because previous studies have shown the presence of beta-glucans in joints of patients with arthritis. For example, the study published by Shiota et al. (2001) [53] measured endotoxins and beta-glucans in the synovial fluid of patients undergoing hemodialysis, osteoarthritis, or rheumatoid arthritis. When analyzing the levels of endotoxin and beta-1,3 or 1,6-glucan, they observed that the concentration of endotoxin was the same in the three groups. However, the concentration of beta-d-glucan was higher in patients on long-term hemodialysis. This finding suggests that beta-d-glucan may have something to do with the pathogenesis of synovitis in hydrarthrosis in long-term hemodialysis patients. Additionally, it is possible that beta-glucan induces the production of cytokines in the synovial fluid of long-term hemodialysis patients. However, in that study, beta-glucan was already present in the joint, probably because it is a fungal infection, which leads to the induction of the disease. This has been reported in other studies, which looked at the presence of serum beta-glucan in Acquired Immune Deficiency Syndrome patients with pneumonia, or respiratory symptoms caused by fungi [54].

In the study by Yoshitomi et al. (2005) [52], they used the injection of zymosan, a crude extract from the yeast cell wall, which can cause severe arthritis in mice. The main constituent of zymosan is the glucose polymer beta-1,3-glucans, which are responsible for the arthritogenic effect. In addition, in this study, they demonstrated that mice do not develop arthritis in a sterile environment and that antifungal treatment can prevent arthritis even in a microbial environment prone to arthritis. However, they concluded that environmental agents, such as fungi and viruses, can trigger autoimmune arthritis in genetically susceptible individuals.

Some more recent studies have evaluated the use of beta-glucan in the face of chronic inflammation and pain caused by the condition of osteoarthritis. Kim et al. (2012) [55] conducted a study with Polycan of *Aureobasidium pullulans* SM-2001, composed mainly of beta-1,3 1,6-glucan, in male rats with osteoarthritis induced by anterior cruciate ligament transection and partial medial meniscectomy. In this study, three doses of the compound were evaluated (85, 42.5, and 21.25 mg/kg), administered orally once a day for 84 days. The improvement in symptoms was observed through changes in the circumference and maximum angle of knee extension, in addition to histopathology of the cartilage, using the Mankin score and increased intracellular proliferation with intraperitoneal injection of bromodeoxyuridine after 12 weeks after Polycan administration. The results they obtained showed that in the three dosages used, beta-glucan led to lower degrees of joint stiffness and histological damage to cartilage compared to controls. In addition, the number of immunoreactive cells increased with all dosages in the treated group, showing that treatment with beta-glucan induced the proliferation of chondrocytes on the joint surface.

A study by Choi et al. (2015) [56] explored the possible synergistic benefits of Polycan (beta-1,3/1,6-glucan extracted from *Aureobasidium pullulans* SM-2001) and calcium lactate-gluconate (Ca-LG) in the proportion of 1: 9, in a model of osteoarthritis induced in rats. This study was carried out with a total of 80 rats, which were randomly divided into eight groups as follows: control groups without induced OA, control OA, treated with diclofenac, treated with Polycan, treated with Ca-LG, and treated with Polycalcium in different doses. The commercial mixture Polycalcium was administered to the animals in the doses of (50, 100, and 200 mg/kg), orally once a day for 28 days from 1 week after surgery. Prior to this study, this same research group carried out a toxicological test of this substance in rats [57]. In this pilot study, the animals received doses (0, 500, 1000, and 2000 mg/kg in a volume of 10 mL/kg) once a day for 14 days to investigate possible side effects. After 14 days of oral treatment, no treatment-related mortality, clinical signs, or changes in body/organ weight were detected. In addition, no significant changes were observed in hematological, biochemical, macroscopic, or histopathological parameters, compared to controls. Demonstrating that beta-1,3/1,6-glucan is non-toxic to mice and is therefore probably safe for clinical use.

Regarding the results obtained with the blend, they observed that OA-related symptoms were inhibited after 28 days of continuous oral treatment with Polycalcium. In addition, chondrocyte proliferation was induced, which were more favorable in the use of the blend than the compounds offered alone (100 mg). Therefore, in this study, the blend of Polycan and Ca-LG had beneficial synergistic effects on the symptoms of osteoarthritis [56].

Clinical studies in this area are scarce, and only a very recent study evaluated the effects of beta-1,3/1,6-glucan in human patients [58]. In this study, Truong et al. evaluated the efficacy and safety of Polycan, in combination with glucosamine in reducing symptoms associated with knee osteoarthritis. This was a double-blind, randomized clinical trial lasting 12 weeks. Patients were screened from the Hanoi Hospital service in Vietnam, and several patient inclusion or exclusion criteria were included for entry into the study. A total of 100 patients were chosen, and the degree of symptoms varied, so they were allocated into groups according to grades 1 to 3 by the Kellgren and Lawrence system. In addition, people were divided into three groups: group A received 16.7 mg of Polycan and 250 mg of glucosamine, group B 16.7 mg of Polycan and 500 mg of glucosamine, and the control group 500 mg of glucosamine per capsule, administered in three capsules once daily for a period of 12 weeks. Improvement assessments were performed using the Western Ontario and McMaster Universities Osteoarthritis Index (WOMAC) questionnaire, as well as the use of rescue drugs (according to data from a patient-reported diary) and other safety indices (body weight, blood pressure, hematological and biochemical markers).

The results of this study demonstrated that there was a reduction in the WOMAC score in the three groups over the treatment period. When compared to each other, there was a greater reduction in the total WOMAC score in Group B compared to the control group. It is important to note that all groups tested, at week 12, had no hematological or biochemical changes, showing that the doses used are safe for people. However, in this study it was shown that patients with mild or moderate osteoarthritis, who used beta-1,3/1,6-glucan or with glucosamine, in general, improved the clinical signs more evident than patients who received only glucosamine.

Few studies have evaluated the beneficial effect of beta-glucans in animal and human models as shown previously and Beynen et al. (2010) [59] were the pioneers and the only ones to carry out a study evaluating the oral administration of beta-glucan in dogs with osteoarthritis. They performed a completely randomized, double-blind, placebo-controlled design with dogs screened at the clinic and remained in their homes throughout the experiment. The 46 animals that participated in the experiment were divided into two groups with 23 dogs each. Both groups received the same food for 8 weeks, the only difference being the addition of 800 ppm MacroGard^®^ before extrusion for the test group. MacroGard^®^ is a commercial preparation of highly purified beta-1,3/1,6-glucans derived from yeast (*Saccharomyces cerevisiae*). To assess the clinical improvement of symptoms, the owners answered an online questionnaire weekly from the beginning of the treatment, which made the evaluation through a scale ranging from 0 to 100. The punctuated signs were activity (vitality), stiffness, swelling joints, lameness, paralysis, and pain. The first response before the start of treatment was used as a baseline, and from that baseline, improvement or not over time was observed.

When compared to baseline values, administration of beta-1,3/1,6-glucans significantly improved activity (vitality) and stiffness change in clinical signs of stiffness. However, there was no difference when comparing the two groups over time.

In pigs, Li et al. (2006) [60] demonstrated that feeding beta-1,3/1,6-glucans reduced plasma concentrations of pro-inflammatory cytokines, IL-6 and TNFα, and increased the concentration of anti-inflammatory cytokine, IL-10. Thus, intake of beta-1,3/1,6-glucans can reduce inflammation in canine osteoarthritis and thus reduce pain. TNFα also stimulates the production of matrix metalloproteinase-3 (MMP-3) by chondrocytes [61]. MMP-3 is involved in the breakdown of collagen molecules in the cartilage matrix. Thus, it can be suggested that the positive effect of beta-1,3/1,6-glucans is caused by the inhibition of collagen degradation in the cartilage matrix associated with a reduction in inflammation and in the sensation of pain.

Beta-glucans can be considered safe, and it is suggested that their use may be beneficial for animals and people with osteoarthritis. However, more clinical trials are needed to evaluate its role in cases that are not induced and without a controlled environment, as occurs in most studies published in the literature to date.

## 5. Beta-Glucans and Inflammatory Bowel Disease

Inflammatory bowel disease (IBD) is an immune-mediated syndrome of the gastrointestinal (GI) tract that causes chronic inflammation, dysbiosis, loss of oral tolerance, loss of intestinal barrier integrity, and malabsorption [62]. The understanding of IBD pathogenesis is complex and must be carefully studied on a molecular and clinical presentation basis considering differences between the species.

IBD in humans presents itself in two distinguished forms, Crohn’s disease (CD) and ulcerative colitis (UC), with the first being a disease affecting mainly the ileum and colon that can also affect other organs of the GI tract like the stomach and duodenum, and the second, predominantly, being an ulcerative and inflammatory disease limited to superficial layers of the gut (mucosa and superficial submucosa) in the colon [63].

IBD in dogs and cats, on the other hand, is diagnosed as an inflammatory disease with histological findings of mucosal inflammation, without distinguishing the location in GI tract, when all other possible causes of enteritis/infiltrates have been investigated and excluded. It is generically named after the predominant cell infiltrate in the mucosa where the inflammation takes place and, according to the ACVIM’s histopathologic guidelines consensus, its severity (e.g., severe lymphoplasmacytic enterocolitis) [63,64].

The differences between humans and dogs/cats remains in the immunologic profile of the disease; while dogs and cats have a mixed activation of Th1 and Th2 lymphocyte subsets and, consequently, a mixed expression of cytokines, CD is specifically associated with Th1 and Th17 lymphocyte and UC with Th2 [63].

Th1 cells initiate cytotoxicity and cell-mediated response and antagonize Th2 cell function. Th2 cells mediate humoral immunity and antagonize Th1 cell function. Finally, Th17 cells are a distinct lineage of proinflammatory CD4+ T lymphocytes that produce IL-17 that triggers and amplifying the inflammatory process [65,66].

Only a few studies in dogs and cats have evaluated the Th17 subset and did not find evidence for the involvement of Th17 signature cytokines in canine or feline inflammatory bowel disease [67,68,69]. The appointed differences are the main explanation why the disease behaves differently in humans versus dogs and cats, meaning overall more severe in humans and less severe in dogs and cats. Moreover, understanding how the treatment of evolving immunomodulatory drugs, specifically regarding pro- and prebiotics, must be carefully investigated.

IBD treatment is based on the use of immunosuppressant drugs like corticosteroids, cyclosporine, preparations of 5-amino-salicylic acid and, eventually, antibiotics as primary line of defense to suppress inflammation in the gut. As a supporting therapy, adequate nutrition and prebiotics intake plays an important role in minimizing food antigen interaction in the gut mucosa deflecting from the loss of oral tolerance process [70] and may also play a role in the regulation of inflammatory process by increasing the T cell subtypes (Treg) able to regulate/control an exacerbated inflammation [71,72,73].

Beta-glucans are a separated group of polysaccharides able to be selectively fermented by gut bacteria to change their composition (diversity and population) and their fermentation products, the short-chain-fatty acids (SCFAs), lactic acid (LA), and branched-chain fatty acids (BCFA), to provide health benefits to the host [74].

The increased production of SCFAs, along with LA inhibits the growth of pathogenic bacteria by reducing the pH of the intestinal microenvironment. Additionally, butyrate is one of the main nutritive compounds for enterocytes and can repair the damages in mucosa, reduce intracellular transcription of NF-κB, attenuate synthesis of proinflammatory cytokines, and induce differentiation of regulatory T cells (induced Tregs) and differentiation of type 2 macrophages (M2), which is an important productor of IL-10 and therefore, important for the immunoregulatory/anti-inflammatory process [75].

According to some authors, SCFAs are considered a class of nutrients that interact with membrane receptors FFA2, FFA3, GPR109a, and Olfr78 expressed uniquely throughout the gut and signal distinct mechanisms like intestinal motility, hormone secretion, maintenance of the epithelial barrier, and immune cell function. This last function supports the emerging research on intestinal health in pathological conditions, like IBD.

Therefore, beta-glucans act not only by modulating the growth of harmful versus beneficial bacteria, but also through production of nutrients that interact with the gut and immune system of the host, specially through gut-associated lymphoid tissue [76], which harbors over 65% of all the immune cells in the body and over 90% of all Ig-producing cells and makes the intestine the largest immune organ [77].

Beta-glucans are molecules extracted from yeast, bacteria, or cereals that may interact with the immune system, up- or down-regulate immune response, and improve health of the host [13]. For a long time, they were strong activators of cellular immunity exerting a pro-inflammatory effect against pathogens and cancer cells [13]. Therefore, its use in diseases like IBD may seem contradictory, since the pathogenesis relies on exaggerated activation of immune system that requires immunosuppressant drugs to be controlled. Although, in the light of the new studies proving its beneficial effect in pro-inflammatory diseases (specially IBD), by controlling them and alleviating symptoms (Table 2), the term “immunomodulator” appears to be more accurate.

The mechanism underlying the process is complex and still to be elucidated and should be first addressed in a non-disease environment. Marchi et al. (2024) elaborated a study where healthy dogs consumed food with 0.0%, 0.07%, 0.14%, and 0.28% of beta-1,3/1,6-glucan for 140 days. The authors observed that Firmicutes phyla increased according to the dosage only up to 0.14%, Actinobacteria decreased over the period in the higher dosage (0.28%), all treatments with reduced the relative abundance of Bacteroidetes. No alterations were observed for the bacteria metabolites (short-chain fatty acids) in feces. Therefore, the analysis of these compound in fecal matter may not provide any conclusions [14].

In a controlled study performed by Zyła et al., (2019) [73], the immunomodulator capability of oat beta-1,3/1,4-glucan, that is, the capability of rebalancing the ratio of pro-/anti-inflammatory cytokines, was very well demonstrated. In this research, the oat beta-1,3/1,4-glucan protective effect was less pronounced in animals with healthy colons and mainly exerted an anti-inflammatory effect in rats with induced IBD. The authors believed that oat beta-glucans supplementation effectiveness is higher when is applied in the course of ongoing inflammation and not in health situations for prophylactic purposes. This concept was also supported by Municio et al., (2013) [78] in a study that demonstrated that the response of human macrophages to beta-glucans depends on the inflammatory milieu.

Differently from Zyla’s study, DSS-induced IBD mice were protected from mucosa damages when they were pretreated with increasing doses of bacterial beta-1,3-glucan in a study published by Lee et al. (2014) [79]. However, the different result may be justified, not only by the difference of glucan origin, but also by the absence of glucan intake control as in Zyla’s study.

A completely different result was found by Heinsbroek et al. (2015) [80] in a very similar experiment on the aggravation of intestinal inflammation in DSS IBD-induced mice pretreated with beta-glucan. Although, the authors concluded that the origin of glucan might have played a central role in this discordance so that in Lee’s experiment a bacterial beta-1,3-glucan purified from *Agrobacterium* spp. was used and in Heinsbroek’s study, *Alcaligenes faecalis* and *Saccharomyces cerevisae* derived beta-glucan was used.

In parallel, importance should be given to the role of gut bacteria in inflammatory situations which configure the main arms for understanding the disease and endorse the use of prebiotics as a supporting therapy. Like previously explained, dysbiosis is a major consequence of IBD by providing continuous antigenic stimulation for GALT, intra epithelial lymphocytes (IEL), and lamina propria lymphocytes (LPL) [81].

According to Matsuoka and Kanai (2015) [72], for many years, the role of the immune system was solely to distinguish the self from the non-self, determine the degree of pathogenicity of microbes, and adjust the response accordingly by tolerating them (in a symbiotic relationship) or not, when there is probability of harm.

However, gut bacteria can, furthermore, interact directly with immune cells. For example, spore forming bacteria can coordinate T cell responses, induce the development of germinal centers in Peyer’s patches and other intestinal lymphoid organs, and increase production of IgA and Th17 cells [72,82]. Therefore, it can be concluded that the control of bacteria population is of major importance for the control of immune responses.

Until this moment, few studies have been found demonstrating the ability of beta-glucan in modulating gut microbiota in humans [83,84,85], piglets [86], rats [87,88,89], and even in cell suspensions [90] and yogurt [91], but none are related to situations of chronic gut inflammation and no studies have been found in dogs and cats.

**Table 2 microorganisms-12-01071-t002:** Experimental studies published about the effects of beta-glucan in IBD patients or IBD disease model.

Reference	Population	Βeta-Glucan Origin/Dose/Frequency	Main Findings	Limitations
Ganda Mall et al. (2018) [92]	Human with CD	Oat-beta-glucan/nonspecific/6 weeks	No significant effects on intestinal permeability, inflammatory/oxidative levels in blood plasma and self-reported health.	Beta-glucan estimated by food frequency questionnaire and 85% of the patients had insufficient dietary fiber intake.
Spagnuolo et al. (2017) [93]	Human with IBD	Yeast-beta-glucan/55 mg a day/4 weeks	Reduction in abdominal pain together with reduction in bloating and flatulence after four weeks of treatment.	No individualization of the treatments: mixture of beta-glucan, inositol and digestive enzymes (Biointol^®)^ + mesalamine.
Chermesh et al. (2007) [94]	Humans with CD after surgery	Undeclared/2.5 g a day/2 years	No effect on postoperative recurrence of GI signs and markers of inflammation.	No individualization of the treatment: use of a symbiotic (Symbiotic 2000^®^).
Segarra et al. (2016) [95]	Dogs with IBD	Undeclared/26 mg kg^−1^/180 days	Significant 1.53-fold decrease (*p* < 0.01) in median overall histologic score. Higher blood concentrations of PON1, and reduced TAC levels.	No individualization of the treatment: use of a mixture of chondroitin sulfate, resistant starch and MOS and small sample size.
Hallert et al. (2003) [96]	Humans with quiescent UC	Not purified oat beta-glucan/60 g of oat bran a day/12 weeks	Increased butyrate fecal concentrations and reduced abdominal pain and reflux.	Small sample size. Placebo composition undeclared.
Rychlik et al. (2013) [97]	Dogs with IBD	Yeast beta-glucan/7 mg kg^−1^/6 weeks	Lowering of CIBDAI values to below 3, improved histopathological parameters, decreased IL-6 levels, increasing IL-10 concentrations and remission periods longer than six months.	
Lee et al. (2014) [79]	Mice with induced IBD	Bacterial beta-glucan/2.5 or 5 mg kg^−1^/2 weeks prior IBD DSS induction	Recovery of colonic architecture, disease score and histological score. Expression of IL-1β, IL-6, and IL-17α were markedly decreased in the colon. Induction of Tregs population. Reverse of the functional defects of NK cells and excessive IgA production.	DSS induced IBD.
Zyła et al. (2019) [73]	Rats with induced IBD	Oat beta-glucan/1% of low or high molecular weight beta-glucan/21 days	Reduction of the number of T lymphocytes in populations of IELs and LPLs. Reduction of percentage of B cells and IL-12, decreased gene expression and secretion of proinflammatory cytokines and other inflammatory signaling molecules in the tissue of rats with induced colitis.High molecular weight is more effective as a specific internal dressing on the inflamed tissue and low molecular weight modulates the immune cells function on the molecular level.	Beta-glucan intake was not controlled. TBNS induced IBD.
Heinsbroek et al. (2015) [80]	Mice with induced IBD	Bacteria, yeast beta-glucan and glucan phosphate/1 mg a day/2 weeks	Increased histopathologic inflammation score, TNF-α and CCL-2 cytokine. Unlike curdlan and zymosan, glucan phosphate–treated mice did not show significant differences with vehicle treatment in any of the measured cytokines and was the only one to increase IL-10 levels.	DSS induced IBD.

CD, Crohn disease; IBD, inflammatory bowel disease; GI, gastrointestinal; PON1, paraxonase-1; TAC, total antioxidant capacity; MOS, mannooligosaccharides; UC, ulcerative colitis; CIBDAI, Canine Inflammatory Bowel Disease Activity Index; IL: interleukin; DSS, dextran sulfate sodium; NK, natural killer cell; IgA, immunoglobulin A; IEL, intraepithelial T lymphocyte; LPL, lamina propria lymphocyte; TBNS, 6-trinitrobenzene sulfonic acid solution; Tregs, regulatory T cell.

Indeed, the lack of studies about the subject represents a major gap in scientific literature about the application of beta-glucan in IBD patients and harbors an important field for further investigation. In this context, the microbiota modulation pathway can only be inferred in a generalized manner considering the prebiotic property of beta-glucans and its indirect effects of supplementation: the increased production of SCFAs, for example.

Finally, it can be concluded that beta-glucan is a prebiotic that may promote several benefits in patients with IBD due to its immunomodulatory properties in a direct or indirect pathway, although more investigation is needed to identify which origin, molecular weight, dose, and in which situations it can applied so the beneficial effect can be achieved in each species diagnosed with IBD.

## 6. Conclusions

Beta-glucans are biological modifiers capable of modulating the immune environment according to the organism’s needs. This review demonstrates their potential extensive applications in skin diseases, orthopedic diseases, and inflammatory bowel disease. Their mechanisms of action extend beyond gut microbiome shifts to direct interactions with immune cells via receptors.

However, the lack of clear information regarding product concentrations, the inclusion of low-quality evidence papers such as case reports, and the heterogeneity and limited number of studies in dogs and cats render any conclusions underpowered.

Future research should prioritize randomized, double-blinded studies using beta-glucans as exclusive treatment. Factors such as the type of beta-glucan, disease stage, dosage, and duration of treatment need careful analysis. While human research may suffer from diet standardization issues that can interfere with results, this is less of a concern with dogs and cats. Therefore, translating results from pets to humans should be encouraged and could enhance knowledge acquisition in the field.

## Figures and Tables

**Figure 1 microorganisms-12-01071-f001:**
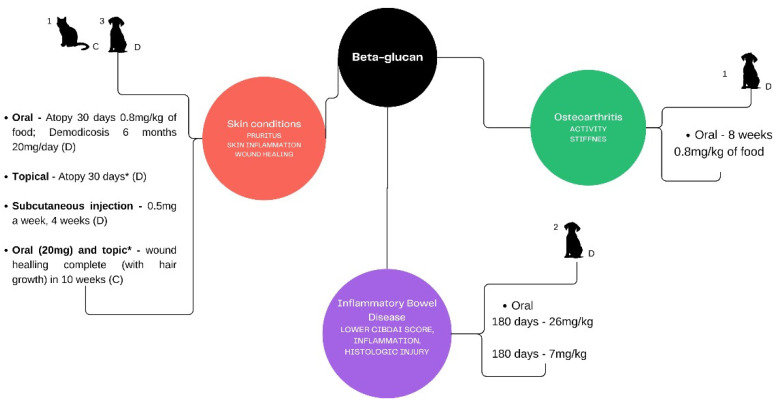
Conceptual diagram of studies or case reports involving beta-glucans, routes of administration, dosages, and treatment length in dogs and cats. Superscript numbers indicate the number of studies found; C = cat; D = dog; * concentration not declared.

**Table 1 microorganisms-12-01071-t001:** Search terms and article selection for beta-glucan research in skin diseases, osteoarthritis, and inflammatory bowel disease.

Subject	Search Terms
Skin disease Articles retrieved: Embase—442 PubMed—1110 Duplicated: 12 Removed (type of paper): 82 Removed (title and abstract): 668 Total: 7	(dog OR canis canis OR canis domesticus OR canis familiaris OR canis lupus familiaris OR dog OR dogs OR cat OR human OR homo sapiens OR human OR human being OR human body OR human race OR human subject OR humans OR man (homo sapiens) OR animal model OR animal disease model OR animal model OR animal models OR model, animal OR models, animal) AND (skin disease OR cutaneous disease OR dermal disease OR dermal disorder OR dermatopathology OR dermatopathy OR dermatosis OR dermopathy OR disease, cutaneous OR disease, dermal OR facial dermatoses OR foot dermatoses OR leg dermatoses OR scalp dermatoses OR skin and connective tissue diseases OR skin disease OR skin diseases OR skin diseases, genetic OR skin disorder) AND (beta-glucan OR beta-dextroglucan OR beta-glucan OR beta-glucans OR beta-glucans OR macrogard)
Osteoarthrites Articles retrieved: Embase—13 PubMed—325 Duplicated: 0 Removed (type of paper): 32 Removed (title and abstract): 298 Total: 8	(dog OR canis canis OR canis domesticus OR canis familiaris OR canis lupus familiaris OR dog OR dogs OR cat OR human OR homo sapiens OR human OR human being OR human body OR human race OR human subject OR humans OR man (homo sapiens) OR animal model/exp OR animal disease model OR animal model OR animal models OR model, animal OR models, animal) AND (osteoarthritis/exp OR arthritis, degenerative OR arthritis, noninflammatory OR arthrosis OR degenerative arthritis OR degenerative joint disease OR noninflammatory arthritis OR osteo-arthritis OR osteo-arthrosis OR osteoarthritis OR osteoarthrosis OR primary osteoarthritis OR rheumatoid arthrosis) AND (beta-glucan/exp OR beta-dextroglucan OR beta-glucan OR beta-glucans OR beta-glucans OR macrogard)
Inflammatory Bowel Disease Articles retrieved: Embase—156 PubMed—72 Duplicated: 6 Removed (type of paper): 13 Removed (title and abstract): 200 Total: 9	(dog OR canis canis OR canis domesticus OR canis familiaris OR canis lupus familiaris OR dog OR dogs OR cat OR felis OR cat OR cats OR feline OR felines OR human OR homo sapiens OR human OR human being OR human body OR human race OR human subject OR humans OR man (homo sapiens) OR animal model OR animal disease model OR animal model OR animal models OR model, animal OR models, animal) AND (beta-glucan OR beta-dextroglucan OR beta-glucan OR beta-glucans OR beta-glucans OR macrogard) AND (inflammatory bowel disease OR inflammatory bowel disease OR inflammatory bowel diseases)

## Data Availability

The original contributions presented in the study are included in the article. Further inquiries can be directed to the corresponding author.

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
