# Peer review of "Translating Human and Animal Model Studies to Dogs’ and Cats’ Veterinary Care: Beta-Glucans Application for Skin Disease, Osteoarthritis, and Inflammatory Bowel Disease Management"

_microorganisms, 2024, doi:10.3390/microorganisms12061071_

Round 1

Reviewer 1 Report

Comments and Suggestions for Authors

Andressa Rodrigues Amaral et al., reviewed the immunomodulatory effects of β-glucans in dogs and cats in positively regulating atopic dermatitis, inflammatory bowel disease, and osteoarthritis. In this review, the authors also discussed the potential mechanisms of β-glucans in different disease. With the growing interests in this research field, this review is timely important by guiding future knowledge applications in clinical practices. I have only 3 minor comments below.

1) Advised to change the term “small animals” in the title to “dogs and cats” to make the scope of the study more specific.

2) Advised to check the second affiliation.

3) Adding 1-2 conceptual diagrams is recommended, as pure text and tables may make readers feel very dull.

Author Response

Dear Editor,

The authors wish to thank you for the attention given to our research. We have considered all the suggestions and comments by the reviewers, which are addressed below. Thank you once again for considering our work for publishing. I hope you are also keeping safe and well.

Kind regards,

Thiago Henrique Annibale Vendramini

Reviewer: 1

Andressa Rodrigues Amaral et al., reviewed the immunomodulatory effects of β-glucans in dogs and cats in positively regulating atopic dermatitis, inflammatory bowel disease, and osteoarthritis. In this review, the authors also discussed the potential mechanisms of β-glucans in different disease. With the growing interests in this research field, this review is timely important by guiding future knowledge applications in clinical practices. I have only 3 minor comments below.

Reviewer: Advised to change the term “small animals” in the title to “dogs and cats” to make the scope of the study more specific.

Author: We made the change, thank you.

Reviewer: Advised to check the second affiliation.

Author: We checked and corrected, thank you.

Reviewer: Adding 1-2 conceptual diagrams is recommended, as pure text and tables may make readers feel very dull.

Author: A conceptual diagram summarizing all studies in dogs and cats was included in at the end of the file.

Reviewer 2 Report

Comments and Suggestions for Authors

Dear Authors,

I read with great interest your article about beta-glucans as possible modulators of inflammatory immune diseases in small animals.

However, there are some aspects that require your attention.

First of all you need to describe the PRISMA protocol used for finding the articles you mention in your manuscript.

In the discussion section regarding skin allergies, you should also mention the use of biologic treatment in this pathology and other pathologies such as allergic rhinitis and sinusitis. Reference this to the article by Cergan R, Berghi ON, Dumitru M, Vrinceanu D, Manole F, Serboiu CS. Biologics for Chronic Rhinosinusitis-A Modern Option for Therapy. Life (Basel). 2023 Nov 5;13(11):2165. doi: 10.3390/life13112165. PMID: 38004305; PMCID: PMC10672088.

You need to insert a clear paragraph about the limitations of the present study, such as possible articles published that were not available open access.

A clear section of final conclusions should be included. And in this section you could mention future developments of the present study such as the translation of the research from animal models into humans.

There are many abbreviations in the text so you need to introduce a list of abbreviations at the end of the article.

Looking forward to receiving the improved version of your manuscript.

Author Response

Dear Editor,

The authors wish to thank you for the attention given to our research. We have considered all the suggestions and comments by the reviewers, which are addressed below. Thank you once again for considering our work for publishing. I hope you are also keeping safe and well.

Kind regards,

Thiago Henrique Annibale Vendramini

Reviewer: 2

Dear Authors,

I read with great interest your article about beta-glucans as possible modulators of inflammatory immune diseases in small animals.

However, there are some aspects that require your attention.

Reviewer: First of all you need to describe the PRISMA protocol used for finding the articles you mention in your manuscript.

Author: A topic “Method of search” and a “Table 1” describing it was included. Thank you.

Reviewer: In the discussion section regarding skin allergies, you should also mention the use of biologic treatment in this pathology and other pathologies such as allergic rhinitis and sinusitis. Reference this to the article by Cergan R, Berghi ON, Dumitru M, Vrinceanu D, Manole F, Serboiu CS. Biologics for Chronic Rhinosinusitis-A Modern Option for Therapy. Life (Basel). 2023 Nov 5;13(11):2165. doi: 10.3390/life13112165. PMID: 38004305; PMCID: PMC10672088.

Author: A paragraph was added considering the information above, as well as referencing the suggested article.

Reviewer: You need to insert a clear paragraph about the limitations of the present study, such as possible articles published that were not available open access.

Author: Thank you. A conclusion section was included. All articles that could provide important information (when selected after systemized search) were accessed. Either through open access or through VPN of the University that holds signature with many journals, or it was specifically requested to the library staff.

Reviewer: A clear section of final conclusions should be included. And in this section you could mention future developments of the present study such as the translation of the research from animal models into humans.

Author: A final topic with conclusion was inserted as requested.

Reviewer: There are many abbreviations in the text so you need to introduce a list of abbreviations at the end of the article.

Author: A list of abbreviations was included at the end.

Reviewer: Looking forward to receiving the improved version of your manuscript.

Author: Thank you for all the suggestions!